# Sports Bra Pressure: Effect on Body Skin Temperature and Wear Comfort

**DOI:** 10.3390/ijerph192315765

**Published:** 2022-11-26

**Authors:** Kit-Lun Yick, Yin-Ching Keung, Annie Yu, Kam-Ho Wong, Kwok-Tung Hui, Joanne Yip

**Affiliations:** 1School of Fashion and Textiles, The Hong Kong Polytechnic University, Hung Hom, Hong Kong; 2Department of Advanced Fibro Science, Kyoto Institute of Technology, Kyoto 606-8585, Japan; 3Laboratory for Artificial Intelligence in Design, Hong Kong Science Park, New Territories, Hong Kong

**Keywords:** bra pressure, skin temperature, treadmill exercise

## Abstract

**Highlights:**

**What are the main findings?**
Shorter underband and shoulder straps significantly increase bra-skin pressure but result in higher positive sensation of bra pressure comfort and breast support during exercise.Increased bra-skin pressure does not significantly change body skin temperature but has significant effects on bra-breast skin temperature during running, cooling down and sitting.

**What is the implication of the main finding?**
High pressures induced by sports bras that habitually considered harmful to the human body may not lead to wear discomfort but enhance bra support sensation and a sense of security to the wearers.

**Abstract:**

Sports bras are an essential apparel for active women, but may exert excessive pressure that negatively affects thermoregulation, thermal comfort and wear sensation. This study measures skin temperature changes during short durations of exercise on a treadmill with different bra pressures. The results based on 21 female subjects (age: 27.2 ± 4.5 years old) show that bras with more pressure at the underband or shoulder straps do not cause statistically significant skin temperature changes during exercise (*p* > 0.05). Nevertheless, compared to the optimal bra fit, significant differences in bra-breast skin temperature are found during running, cooling down and sitting when the bra pressure is increased (*p* < 0.05), particularly under bra cup (T1) in this study. The FLIR thermal images can visualize the skin temperature changes at abdomen throughout the four activity stages. Subjective sensations of bra thermal comfort, pressure and breast support are assessed. Despite the increased pressure on the shoulders and chest wall, perceptions towards thermal comfort remain unchanged. The perceived pressure comfort and support sensation amongst the 4 bra conditions are comparable. Interestingly, positive sensations of pressure comfort and breast support are perceived with a tight-fitting sports bra during treadmill exercise. High pressures induced by sports bras (>4 kPa) that habitually considered harmful to the human body may not lead to wear discomfort but enhance bra support sensation and a sense of security to the wearers.

## 1. Introduction

With growing support for a healthy lifestyle, sports apparel and activewear have become an important segment of the global apparel market. Incorporating new concepts in product design and material, activewear is now generally designed to allow free movement to enhance sporting performance and protect the wearer from sports-related injuries [1,2]. For example, sports bras are designed to support the soft tissues of the breasts and control breast motion exacerbated during physical activities due to the absence of anatomical support of the breasts [3,4]. However, sports bras which exert high compressive forces on the chest may initiate other physiological and psychological effects on the body such as poor blood flow and wear discomfort [5,6]. Parameters for wear comfort yet have not been verified and pressure evaluations of bras have not been done in the literature. For these reasons, the problems of excessive breast motion and wear discomfort still affect most females and even prevent them from participating in physical activities [7]. More systematic research that investigates the pressure induced by the different components of a sports bra, such as shoulder straps and underband, as well as the perceived pressure sensation of the wearers are therefore necessary. The findings can provide reliable information to bra designers to improve the sports bra designs with adequate and comfortable pressures so that women can engage in more physical activities. 

A sports bra that provides adequate compression to control breast motion and protect the breasts from injury, but also offers a sense of support and security is critical [8]. Nevertheless, excessive local skin pressure can harm the skin and hinder body movement, which are also considered harmful to the body because it leads to chronic constipation and diarrhea, headaches, and even visceral displacement [9]. Lee, Hyun and Tokura [10] indicated that the pressure exerted by bras and girdles onto the skin could suppress the nocturnal elevation of salivary melatonin, thus resulting in an increase of the rectal temperature. The increase can cause health issues such as organ failure. In designing sports bras, the shoulder strap and bra band are usually the critical areas for evaluating the bra pressure and wear comfort, which are mainly determined by the elastic modulus of the elastic tape [11,12]. Overly tight straps not only cause pressure discomfort, but potentially shoulder and neck pain and headaches [13]. Wider shoulder straps with increased contact surface enable the even distribution of pressure. As suggested by Van Jonsson [14], the bra underband which bears over 80% of the weight of the breasts should be fully stretched to resist the downward gravity forces of the breasts. Umemoto et al. [15] measured the interface pressure between different bras and the body. They found that the highest tolerated pressure is 3.2 kPa for the shoulder strap and 1.47–2.13 kPa for the underband [15]. When shoulder strap orientations are compared, a high level of pressure (>4 kPa) from a sports bra with cross-back shoulder straps may be intolerable [11]. Nevertheless, the mean pressure that is imposed onto the body from a bra greatly varies with anatomic location, body fat, body movement and posture which lead to different tactile and pressure sensations [16]. Apart from pressure asserted by the straps and underband which has been investigated in previous studies, the influence of bra pressure on thermal and moisture comfort, and even overall psycho-physiological response have been generally neglected [9].

Discomfort caused by a sports bra is not merely caused by the bra pressure, since thermal discomfort associated with a tightly fitting sports bra is frequently reported. In assessing compression garments, body core and skin temperature are therefore used to measure the thermoregulatory behaviour of the human body [17,18]. During physical activity, the increases in the blood flow to the skin cause body heat which is transferred from the deep body tissues to the skin and environment [19]. Wearing high compression sports bras, however, inhibits heat dissipation and reduces the exchange of air beneath the clothing with the environment. As compared to the bare breasts, wearing a bra can negatively affect thermoregulation and the ability of the skin of the breasts to cool down, as well as thermal comfort following exercise [20]. The increase of skin temperature from the strain of exercise and perspiration is influenced by the type of garment, properties of the fabrication materials and the associated wear pressure and pressure discomfort [9,16]. As conventional bras made of polyurethane cups result in increased core body temperature during physical activities, Leung et al. [17] proposed a heat-reduction mastectomy bra with a perforated structure to facilitate heat dissipation, thus improving thermal comfort with reduction of body temperature. Alternatives also include the use of fabrics with dynamic moisture properties and highly breathable fabrics for increasing thermal comfort [21]. However, there is little published information on sports bra pressure evaluation and its influence on the physiological response and skin temperature change of the wearers. Little research has been carried out to measure the pressure comfort and bra support sensations of sports bras. 

Although sports bras have been advocated as a necessary apparel item for active women, the pressure effect of a sports bra on skin temperature and thermal comfort has yet to be investigated. Changes in physiological responses such as the body skin temperature and comfort sensation towards bra pressure induced by a tightly fitting sports bra, will determine whether thermoregulation factors are an important consideration in sports bra design [9,16,22,23]. The objective of this study is to measure the skin temperature changes caused by increased bra pressures at the underband and shoulder straps during a short duration of treadmill running. Subjective perceived sensation of thermal comfort, pressure together with overall support are assessed after running trials for each of the four bra conditions studied. We hypothesize that the different bra pressures may lead to differences in temperature changes of the body as well as perceived bra comfort with physical activity. The findings around the bra pressure and its influence on the physiological response of the wearers provide bra designers with insight into the parameters of sports bra design with optimal control of compression, thus enhancing wear comfort and the confidence of women to engage in physical activities.

## 2. Materials and Methods

This study adopts a within-subject repeated-measures design. The participants were instructed to wear four different bras with different tensions and materials during the wear trials. The skin temperature under the bra cup and elastic underband, as well as the exposed skin on the abdomen were measured during a short duration of treadmill running. Subjectively perceived comfort with respect to thermal heat, pressure and the overall support of the bras was recorded after the running trials.

### 2.1. Participants 

A total of 21 healthy young women between 19 and 35 years old (mean = 27.2, S.D. = 4.5) were recruited for a treadmill running experiment in the various bra conditions. They are all Chinese, with an average height of 160.2 cm (S.D. = 5.6 cm) and average weight of 55.7 kg (S.D. = 5.4 kg). Their mean bust circumference is 88.1 cm (S.D. = 3.2 cm) and the mean underbust circumference is 74.9 cm (S.D. = 2.9 cm). Their bra band size ranges from 70 to 80, while their cup size ranges from A to D. Given that breast surgery could affect the natural breast shape, those who have had breast surgery were excluded. Ethics approval (ref. NO. HSEARS20210305003) was obtained from the Human Ethics Committee of the Hong Kong Polytechnic University. The subjects were informed of the experimental procedure, possible consequences, and the purpose of collecting the experimental data. Written informed consent was obtained from the subjects before they participated in the experiment.

### 2.2. Experimental Conditions

A changeable sports bra (Figure 1) that allows adjustment of tension or replacement of the shoulder straps in a flexible manner was used for the experiment [23]. The length of the shoulder straps and bra band could also be adjusted by using the sliders for each bra condition so that the effect of the bra tension and pressure on the thermal responses can be systematically evaluated. 

Based on the length adjustments of the shoulder strap and bra band, a total of 4 different bra conditions were created for the experiment, which include: the optimum fit (Bra A), a 15% length reduction of the bra band (Bra B), a 15% length reduction of the shoulder straps (Bra C), and a cushioning shoulder strap made of foam material (Bra D). With the help of a professional bra fitter, the optimal lengths of the band and the shoulder strap based on a fit trial were identified for a tailored fit for each of the participants. A summary of the four bra conditions is presented in Table 1.

### 2.3. Experimental Protocol

The experiment was carried out in a conditioning room from 12:00 to 16:00 in ambient conditions at a temperature of 22 ± 1 °C and humidity of 70 ± 5%. During the experiment, all of the participants wore standard cotton shorts. The participants were required to run for 15 min on a treadmill at 8 km/h in each bra condition. The Novel Pliance-X pressure system, which has been validated for measuring the interfacial pressure induced by garments onto the human body, was used to measure the bra-skin pressure by placing the sensor between the skin and the bra components [24,25]. The strip sensor used was 10 mm in diameter (in a contact area of 78.54 mm^2^) and 0.95 mm in thickness. It is a capacitive sensor with a sensing pressure range from 0.5 kPa to 60 kPa and the experimental error has been proven to be less than 0.13 kPa. The sensor has high linearity between the applied pressure and sensor outputs and good repeatability with coefficients of variations less than 0.1 [26]. The bra pressure at five locations were measured, including the underband, upper band, shoulder strap, side seam and armhole, before and after exercise (see Figure 2).

The skin temperature of three measured points including under the bra cup, under the elastic woven band at the back of the subject; and the exposed skin on the abdomen (see Figure 2 and Figure 3) was recorded continuously by using wireless temperature sensors, namely iButtons (Thermocrons HC, OnSolution). The sensors have been validated for measuring the surface temperature of skin where were checked against a validated mercury thermometer. The measurement was taken at intervals of every 30 s during treadmill running. With reference to the age-predicted maximal heart rate, the heart rate of each subject was continuously monitored throughout the experiment by using a heart rate monitor (Polar, OH1 optical heart rate sensor). The subjectively perceived comfort in terms of thermal, pressure and the overall support sensation towards each bra condition was recorded by using a visual analogue scale (VAS) based on a Likert scale of 1 (negative) to 10 (positive). Thermal images of the participants were taken by using a digital infrared camera (FLIR E95, 464 × 348 pixels, thermal sensitivity of 0.03 °C). All the measurements used for the evaluation of body skin temperature were taken in the bare breast condition.

To reduce fatigue and ensure full recovery, the participants completed one bra condition each time, so that all 4 bra conditions were completed within a period of 8–12 days. Each session was conducted at the same time of the day to avoid variation due to the circadian rhythm of the body temperature. Prior to each bra condition, the participants were instructed to rest and sit in a relaxed position for around 15 min to stabilize their core temperature T_c_ and acclimatize (Figure 4). Then, the participants were required to walk at a speed of 5 km/h for 5 min as a warmup. After that, they ran for 15 min at a speed of 8 km/h, followed by walking slowly at a speed of 3 km/h for 10 min as active recovery. Finally, 15 min was given for resting in a sitting position as passive recovery, so that each bra experiment was 60 min in total. The treadmill was set at zero inclination.

### 2.4. Subjective Sensation Rating 

Subjective perceived pressure, support and thermal comfort are measured after the treadmill test for each of the four bra tension settings (see Table 2). After each treadmill test, the participants were instructed to rate the level of pressure, support, and thermal comfort of the bra. The perceived sensations were rated by using a 10-point Likert-like rating scale. The subjective ratings on the thermal comfort and pressure comfort of the 4 bra conditions were compared. The relationships among the objective data including the skin surface temperature obtained by using the iButtons and the bra-skin pressure measured by using the strip sensor, respectively, and the subjective perception of the bras are investigated.

### 2.5. Statistical Analysis 

The statistical analysis was processed by using the Statistical Package for the Social Sciences program (SPSS^®^21, IBM^®^ Corporation, New York, NY, USA). A general linear analysis was conducted to evaluate the effect of the bra conditions on the changes in the skin temperature of the different body regions. 

A repeated-measures analysis of variance (ANOVA) was conducted on all of the bra pressure and body skin temperature measurements to evaluate the within-subject effect of (1) the 4 bra conditions and (2) before and after exercise. Pearson’s correlation coefficients were used to evaluate the relationship of increased bra pressure between the skin temperature and subjective sensation upon treadmill running. A series of independent sample T-tests were conducted to compare the magnitude of the bra pressure, skin temperatures obtained from different regions of the body and subjective sensations. The level of significance was set to 0.05. 

## 3. Results

### 3.1. Bra Pressure

The statistical results of 21 subjects on pressure values at the 5 measured locations; namely, the underband, upperband, shoulder strap, side seam and armhole, are presented in Table 3 and Figure 5. The highest mean pressure is found at the shoulders for Bra Condition C (before exercise: 4.66 kPa; after exercise: 4.41 kPa); and side seam in Bra Condition B (before exercise: 4.25 kPa; after exercise: 4.48 kPa) with reduced lengths of the shoulder strap and band. The length reduction of the band in Bra Condition B shows a significant pressure increase at the underband, upperband, side seam and armhole. On the other hand, the length reduction of the shoulder straps in Bra Condition C leads to a significant pressure increase at the shoulders, while the high level of pressure at the shoulder can be reduced by using the cushioning shoulder strap made of foam material (Bra D).

The statistical results showed a significant interaction effect between (i) the 5 measured locations (*p* < 0.05) and (ii) the 4 bra conditions (*p* < 0.05), and pressure values exerted on the body both before and after exercising on the treadmill.

### 3.2. Changes in Body Skin Temperature

#### 3.2.1. Skin Temperature Changes during 4 Stages of Treadmill Exercise

Figure 6 shows the average change in skin temperature of the 21 subjects with the 45 min of treadmill exercise (rest and relax for 15 min (Stage 0), 5 min walk as a warmup (Stage I), 15 min of running (Stage II), 10 min of slow walking (Stage III) and 15 min of sitting (Stage IV)) after 10 min of sitting at (a) T1 (right breast); (b) T2 (abdomen); and (c) T3 (mid-back). At T1 and T3, all of the bra conditions result in a slight increase in skin temperature at Stage I, followed by a gradual increase at T1 and a slight decrease at T3 in Stage II. At T2, all of the bra conditions show a continuous gradual reduction in skin temperature from the start until the end of Stage II. All of the bra conditions then show a sharp increase in skin temperature at the end of Stage II, followed by a slight increase then gradual decline until the end of the sitting at T1 and T3, and a slow gradual increase at T2. Nevertheless, the changes in skin temperatures during the treadmill exercise with time in this study are not statistically significant (*p* > 0.05).

#### 3.2.2. Skin Temperature Changes during 4 Stages of Treadmill Exercise

The differences in skin temperature at T1, T2 and T3 are not statistically significant amongst the 4 bra conditions during the treadmill exercise (*p* > 0.05). Figure 7 shows the rate of change in skin temperature at T1 and T2, respectively, which points to the difference between the temperatures at the start and end of each stage. T1 and T3 show a slight increase from Stage 0 to I as the body starts to activate vasodilation and body heat starts to radiate, while T2 shows a slight decrease as it is exposed to the external environment. From Stages I to II, a significant temperature change can be observed at T1, T2 and T3. At T1, Bra Conditions A and C show a significant increase in skin temperature. At T2, all of the conditions show a significant decline in skin temperature during running, in which Bra Condition D has the highest decline. At T3, different temperature changes are found for all of the bra conditions, with Bra Conditions A and C showing a decrease and B and D showing an increase. At the end of Stage III, T1 shows a decrease while T2 and T3 show a similar rate of increase. At Stage IV, T1 and T3 show a similar rate of decline while T2 shows an increase in the skin temperature.

#### 3.2.3. Skin Temperature of 4 Bra Conditions in Each Activity Stage

The skin temperature obtained from Bra Conditions B, C and D are also paired with Bra Condition A (base) at each activity phase, and the statistical results are presented in Table 4. At T1, the pairwise comparison shows that the tension settings of Bra Conditions B, C and D have statistically significant effects on the skin temperature (*p* < 0.05 as highlighted) when compared with Bra Condition A in Stages II to IV. The skin temperature changes among the bra conditions at T2 do not show any statistically differences for any of the stages (*p >* 0.05). At T3, Bra Condition D has a significant effect on the skin temperature in Stages 0 and I, but no significant change in skin temperature was found in other stages as compared to Bra Condition A (*p >* 0.05).

#### 3.2.4. Body Skin Temperature Distribution from FLIR Images

There is no significant differences found in the skin temperature amongst the 4 bra conditions between the measurements obtained by using iButtons and infrared thermography (*p >* 0.05). However, a large gap difference was found between measurements from iButtons and the FLIR thermal images. Analyses of skin temperature measured with these two methods show that they are not comparable. As there is no guide that indicates the best tool for obtaining skin temperature [27], the skin temperature from both methods is analysed separately. The FLIR thermal images detect the surface temperature that could show the difference in the temperature measured with iButtons which are used in direct contact with the skin to record the skin temperature. Therefore, the FLIR thermal images can provide additional information on the thermal analysis. Figure 8 shows the FLIR images obtained during the four stages with the 4 bra conditions in the frontal view. From the images obtained after Stage III, a temperature decline at T2 is obvious as shown by the change in colour. A temperature change at T1 and T3 is not obvious in the images as the two areas are covered by the bra and there is only a slight change in colour. 

### 3.3. Subjective Sensation Rating

There are no statistical differences among the bra conditions in terms of subjective rating of the thermal comfort of the bras (*p >* 0.05). No significance can be found among the measured skin temperatures and the subjectively perceived thermal heat (*p >* 0.05). Table 5 shows that the bras with various tension levels do not have notable effects on the subjectively perceived thermal comfort. Perhaps the changed tension of the bra components is too small for the wearer to sense the temperature change during motion.

Subjective sensation of pressure and support of the bras do not show a significant relationship with the measured bra pressures. The mean pressure ratings persistently range from 7.00–7.24 for the 4 bra conditions. A similar trend is also found for support sensation, with an average rating of 7.99 regardless of the increase in bra pressure. Note that Bra Condition B has the highest ratings in pressure comfort and support sensation (5 and 6), respectively, as compared to the other bra conditions with lower ratings of 2 or 3.

## 4. Discussion

Sports bras are anatomically engineered to offer breast support and reduce exercise-induced breast discomfort and pain during physical activities. Proper bra pressure not only helps to secure the breasts in place, but also provides a sense of support and security, thus facilitating free and safe body movement and increasing exercise efficiency [10]. Despite the increasing advancements of sports bras, there is still inherent ambiguity around bra pressure due to differences in breast size, anatomic location, amount of body fat and body posture. Even though garment pressure that exceeds 4 kPa is considered harmful to the body, no specific guidelines have been developed for designing sports bras in relation to a comfortable range of pressure for breast support. The influence of the intensity of pressure on the physiological response of the human body during exercise and the associated pressure and comfort sensations has not been fully investigated. Breast discomfort and complaints of bra pressure are therefore frequently reported, which become a major barrier to physical activity, particularly for women with large breasts [3]. On the other hand, to avoid breast injury such as contact breast injury during football activities, female athletes wear breast protective paddings and even an additional sport bra [28], thus resulting in discomfort from heat and perspiration which negatively affect the physiological response of the wearer and her sporting performance. This study, therefore, measures the bra-skin interface pressure of 4 different bra conditions by using a changeable sports bra design so that the immediate changes in body skin temperature in different bra pressure conditions during treadmill running can be recorded and compared. Our findings are not fully in line with our hypothesis that as opposed to the optimal pressure condition, there is no significant difference in body skin temperature with increased bra pressure. When pressure at the underband and shoulder straps is increased, statistically significant skin temperature differences are found under the bra cup during running, cooling down and sitting. Surprisingly, the tightly fitting sports bra condition is received in a positive manner during treadmill exercise. 

### 4.1. Bra Pressures

The significant interaction between the T1, T2 and T3 and Bra Conditions A, B, C and D indicates the effectiveness of the experimental setting. The 4 tension levels of the bra conditions have a significant effect on the pressure exerted onto the body at various locations of the sports bra before and after the treadmill exercise. The mean pressure at the side seam was found to be the highest with Bra Condition B when the underband is reduced in length by 15%; while the mean pressure at the shoulder is the highest with Bra Condition C when the shoulder strap s reduced in length by 15% (4.66 kPa). It is commonly reported that bras with excessive pressure (2.1 kPa or higher at the underband and 3.2 kPa or above at the shoulder straps) cause pressure discomfort, or even contribute to negative health problems in previous studies [10,13]. High pressure generated at the interface between the bra strap and shoulders leads to pressure discomfort and deep furrows in the soft tissues, thus potentially causing shoulder and neck pain and headaches. The remedy could be cushioning shoulder straps to effectively reduce pressure. Thus, Bra Condition D with cushioned shoulder straps show a lower mean pressure when compared to the other bra conditions.

Note that Bra Condition C which has the highest mean pressure at the shoulder strap does not show an impact on the subjective rating of pressure comfort. On the other hand, Bra Condition B with the highest mean pressure at the underband (3.07 kPa) has a higher rating of pressure comfort and support sensation. According to Liu et al. [12], the underband is the bra component that provides the most support in a bra, so the underband pressure has a major effect on subjectively perceived comfort. During the test, Bra Condition B was considered to be “overly tight” with an underband reduced in length by 15% when compared with the optimum fit. The subjective test however shows positive feedback towards the pressure comfort and level of support. This result may be explained by the fact that a high proportion of women have poor ability to independently choose a well-fitting bra as their own daily bra [29]. Even previous studies that have carried out a mathematical analysis on the pressure comfort of the bra underband [12] indicate that perceived tightness is subjective and can greatly vary. A tight underband exerts a high level of pressure, but the wear comfort level might not change along with the pressure level, even if it is not considered a good “fit” in objective tests of fit. The relationship between bra tension level, pressure and wear comfort of sports bra needs to be investigated more in depth. Besides the underband, Bra Condition B also shows an increase in pressure at the upperband, side seam and armhole (2.91 to 4.24 kPa). As the band is the primary support for the breasts, the width and stretch must be constant so that the band sits firmly and comfortably around the chest. Bra components are usually reviewed individually when carrying out fitting tests, and there are no similar studies that show the relationship among bra components. Due to the structure of a bra, reducing the width of the band causes structural changes to the bra during the fitting process, such as shape of the armhole and position of the side seam. This suggests that the appropriate tension control of the bra band is vital because it affects the overall bra-skin interface, and at the same time, can effectively and securely hold a bra in place during physical activities.

### 4.2. Changes in Body Skin Temperature

By observing the overall trend of the temperature change for Stages 0 to IV, the peak skin temperature at T1 and T3 occurs at Stage III, which may be due to the increased thermal insulation of the sports bra which has an extra layer that covers these 2 points. During running, the body generates and exchanges heat with the environment through heat loss. As T1 and T3 are under the bra layers, heat is trapped inside during treadmill running and radiates at a lower rate than the generation of body heat during dynamic motion. At T2, all of the bra conditions show a continuous gradual decline from the start until the end of Stage II. This decline may be due to exposure of T2 to the environment and blood mainly flows to body parts with higher muscle intensity. Among the 4 bra conditions, even though a significant increase of pressure is applied by reducing the length of the shoulder strap and underband, the change in the bra tension setting did not have a significant impact on the skin temperature. The limited contact of the bra with the body might be the reason for this phenomenon.

During Stages I, II and III, a significant relationship is found between Bra Condition A and Bra Conditions B, C and D at T1. As Bra Condition A is the optimum bra fit while Bra Conditions B and C have a tighter shoulder strap and underband, respectively, the air gap between the pad of the bra layer and breast is believed to be smaller. A tighter shoulder strap pulls the pad of the bra towards the chest, while a tighter underband pulls the pad of the bra towards the lower bust and remains there firmly against the underbust area. In Stage III, the upper torso movement increases the contact between the breast and pad of the bra. Moreover, more heat is trapped in the smaller air gap under the pad of the bra. Therefore, the increased tension of the bra components may be the reason for the differences at T1. 

Even though the skin temperature measurements with the use of iButtons and infrared thermography are not comparable [29], there is no significant difference found in this study (*p >* 0.05). Since each iButton can only measure the skin temperature in a small area of the body of 16.3 mm in diameter, the set of thermal images provides additional information of the skin temperature distribution of specific body regions in the different bra conditions.

### 4.3. Thermal and Comfort Sensations

In terms of the perceived comfort sensation with thermal heat and pressure, the subjective ratings do not fully align with the skin temperature and the bra pressure measurements. It is interesting to see that the length reduction of the bra band does not have a negative influence on the perceived pressure comfort sensation and the overall support sensation. In contrast, the increased shoulder strap pressure results in the lowest mean rating in pressure comfort sensation amongst the bra conditions studied. The increased bra pressure with reduced bra band and shoulder strap lengths also results in slightly lower mean ratings of thermal comfort sensation as compared to the other bra samples. Note that variations in thermal and pressure comfort sensitivities amongst the subjects and body parts are remarkable [30,31]. 

As shown in this study, the bony joint of the shoulder tolerates higher pressure values than the band circumference region. It is also noted that pressure comfort is influenced by wear time in that the pressure receptors can adapt to stimuli rapidly and become less responsive with time towards external pressure stimulation [32].

Note that the findings should be considered within the context of the study limitations. First, the sample size is comparably small. Only 21 young female participants who are in the 19 to 35 years old cohort volunteered to take part in the study and all of them are Chinese who are residing in Hong Kong. Therefore, the results of this study should be interpreted with care since the bra size of the female participants are relatively small based on their underbust circumference. Women with larger breasts generally experience increased pain and pressure generated at the interface between the bra strap and shoulders, which may cause deep furrows in the soft tissues. Future studies with a larger population who have larger breasts are recommended for validating and generalizing the results. A braless condition can also be considered in future studies so that the influence of pressure from the sports bras on physiological responses and changes in body skin temperatures with various bra conditions can be investigated to avoid the impact of external pressure on the measurement. Additionally, a compression sports bra style is used in this study, but future studies could use other types of sports bras made with different design features and material properties. With reference to previous studies, the effects of the bra pressure on the heart rate, skin blood flow, respiratory functions, perspiration, metabolism, etc. can also be considered to better understand the intensity of bra pressure on the physiological responses of the body during exercise [16,33]. Nevertheless, this study provides preliminary evidence that addresses the influence of bra pressure on body skin temperature and comfort sensation with physical activity, thereby providing the basis to advance sports bra designs so that there is adequate pressure and wear comfort.

## 5. Conclusions

Despite the increasing demand for sports bras, knowledge of bra-skin interface pressure, comfort sensations and their influence on the physiological responses of the female body during sports activities is deficient. The study confirms that the increased pressure induced by a tightly fitting bra band and shoulder straps does not result in significant changes in the skin temperature throughout exercising on a treadmill. The increased pressure intensity only shows significant effects on skin temperature changes under the bra cup during Stages II to IV in this study. It is interesting that a high bra pressure of 4 kPa or higher at the side seam, bra band and shoulder strap are associated with positive subjective sensations of pressure comfort and support when they are worn during treadmill exercise. The findings of this study provide scientific evidence of body responses caused by the high pressure of bras, hence advancing the design and development of sports bras to improve pressure intensity and wear comfort.

## Figures and Tables

**Figure 1 ijerph-19-15765-f001:**
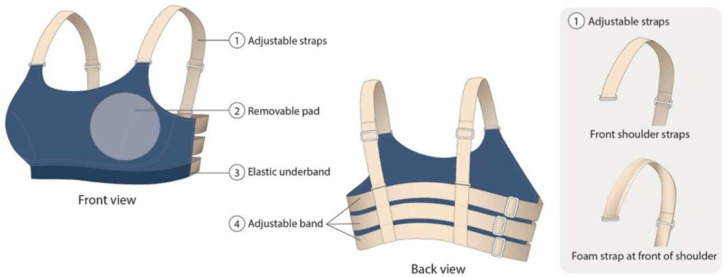
Structure of changeable sports bra.

**Figure 2 ijerph-19-15765-f002:**
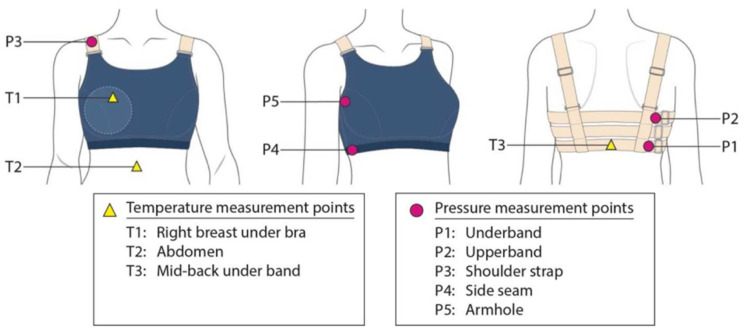
Location of pressure and temperature sensors.

**Figure 3 ijerph-19-15765-f003:**
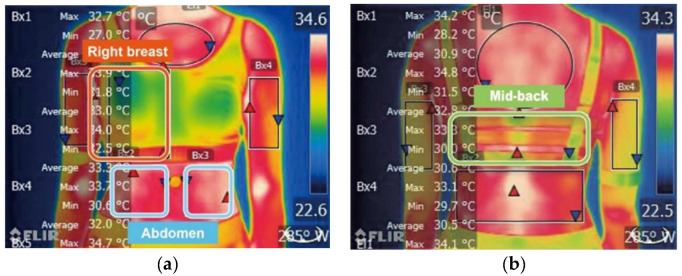
Skin temperature (FLIR) locations with regions of interest (ROIs) at (**a**) Front view, right breast and abdomen; and (**b**) Back view, and mid-back.

**Figure 4 ijerph-19-15765-f004:**
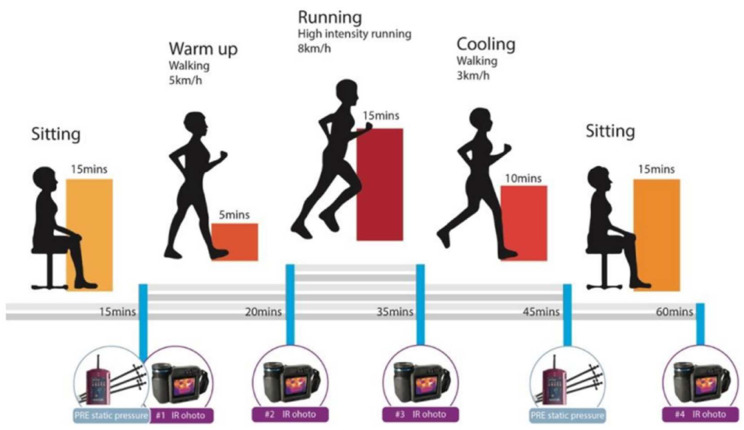
Protocol design of treadmill exercise session.

**Figure 5 ijerph-19-15765-f005:**
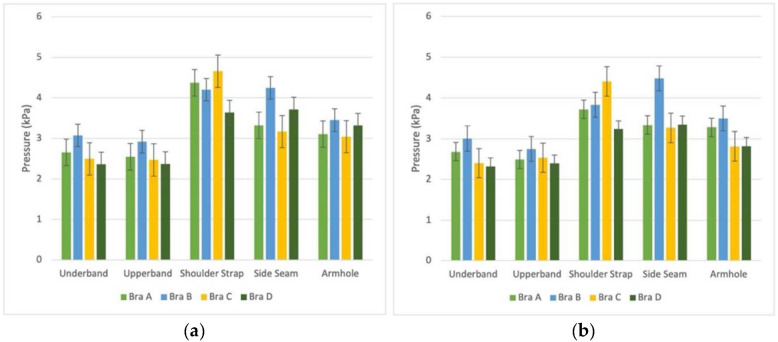
Pressure values at 5 locations (**a**) before treadmill exercise and (**b**) after treadmill exercise.

**Figure 6 ijerph-19-15765-f006:**
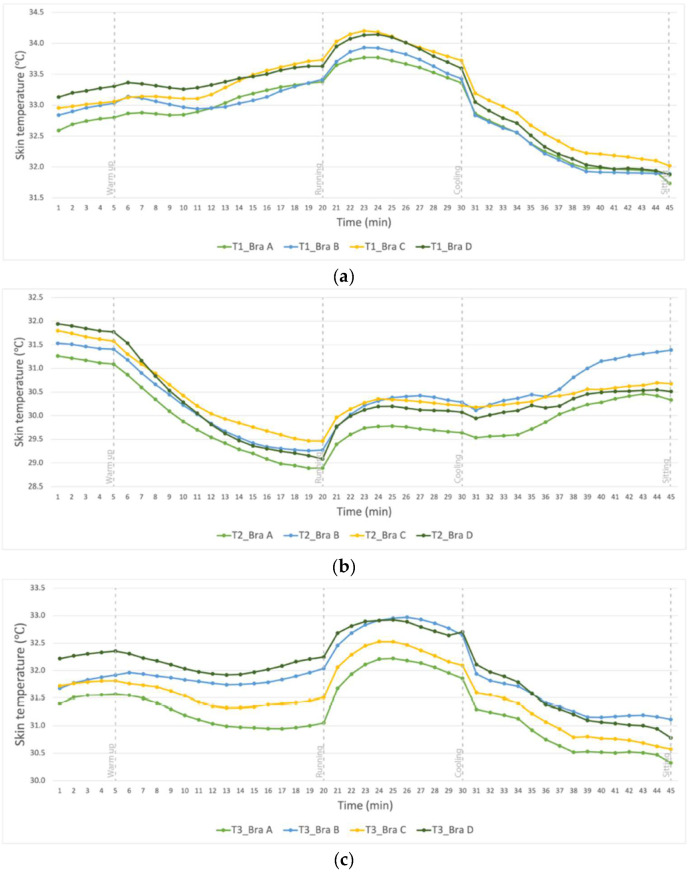
Changes in skin temperature in Stages 0 to IV of treadmill exercise at (**a**) right breast (T1) (**b**) abdomen (T2) and (**c**) mid-back (T3).

**Figure 7 ijerph-19-15765-f007:**
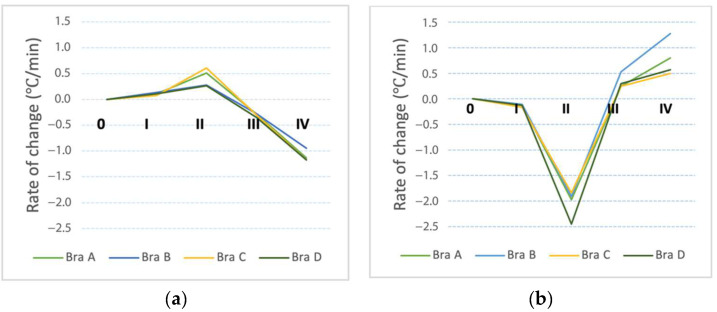
Rate of change in skin temperature at: (**a**) right breast (T1) and (**b**) abdomen (T2) during treadmill exercise.

**Figure 8 ijerph-19-15765-f008:**
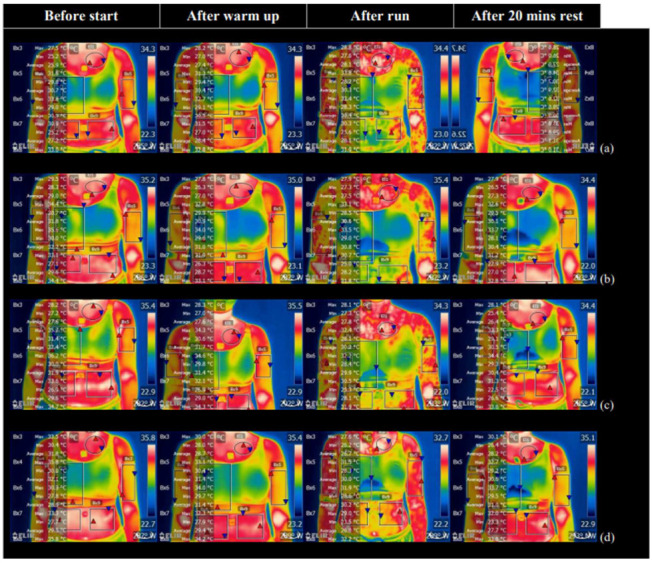
FLIR images for Stages I to II: Bra Conditions (**a**) A; (**b**) B; (**c**) C; and (**d**) D in frontal view.

**Table 1 ijerph-19-15765-t001:** Fabrication details and fit adjustments of the 4 bra conditions.

	Bra A	Bra B	Bra C	Bra D
Front panel structure	Outer shell: Single jersey knitted fabric (55% polyamide & 23% elastane) Cup: polyurethane foam cup in thickness of 2 mm Liner: Powernet mesh (100% nylon)			
Band structure	3 pieces of elastic woven tape with width of 25 mm, Thickness of 1.31 mm, and Young’s Modulus of 0.68 MPa			
Shoulder strap structure	Elastic woven tape with width of 25 mm, thickness of 1.31 mm, and Young’s Modulus of 0.68 MPa			Foam straps with width of 25 mm, thickness of 3.21 mm, and Young’s Modulus of 0.70 MPa
Band Length *	Optimum Fit	Tight Fit (15% length reduction of band)	Optimum Fit	Optimum Fit
Shoulder Strap Length #	Optimum Fit	Optimum Fit	Tight Fit (15% length reduction of shoulder straps)	Optimum Fit

Notes: * The length of the bra bands is adjusted for each subject based on their band circumference measurement; # Shoulder strap lengths are adjusted for each subject based on a measurement from the apex to upperband.

**Table 2 ijerph-19-15765-t002:** Subjective sensation rating.

1	Thermal sensation
	Lowest thermal comfort	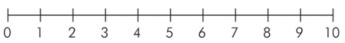	Highest thermal comfort
2	Pressure sensation
	Lowest pressure comfort	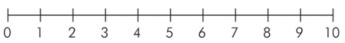	Highest pressure comfort
3	Bra support sensation
	Notsupportive	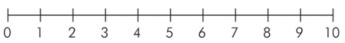	Mostsupportive

**Table 3 ijerph-19-15765-t003:** Summary of descriptive statistics of bra pressure before and after exercising on treadmill.

Location	Bra Condition	Bra Pressure before Exercise (kPa)			Bra Pressure after Exercise (kPa)		
		Mean	Std. Deviation	N	Mean	Std. Deviation	N
Underband	A	2.66	0.81	21	2.68	0.63	21
	B	3.07	0.95	21	3.01	0.67	21
	C	2.50	0.51	21	2.40	0.55	21
	D	2.36	0.32	21	2.32	0.24	21
Upperband	A	2.55	0.55	21	2.49	0.36	21
	B	2.92	0.68	21	2.75	0.51	21
	C	2.47	0.42	21	2.53	0.57	21
	D	2.37	0.32	21	2.39	0.38	21
Shoulder strap	A	4.37	1.49	21	3.72	1.11	21
	B	4.20	1.41	21	3.83	1.34	21
	C	4.66	1.30	21	4.41	1.50	21
	D	3.64	1.17	21	3.23	1.25	21
Side seam	A	3.32	1.02	21	3.34	1.30	21
	B	4.25	1.46	21	4.48	1.40	21
	C	3.17	0.92	21	3.27	0.97	21
	D	3.71	1.63	21	3.35	1.25	21
Armhole	A	3.11	1.02	21	3.28	1.12	21
	B	3.45	1.09	21	3.50	1.17	21
	C	3.04	1.05	21	2.81	0.79	21
	D	3.32	1.52	21	2.82	0.94	21

**Table 4 ijerph-19-15765-t004:** Statistical results of skin temperature at T1, T2 and T3 in Stages 0 to IV.

Stage	Bra Condition (Base)	Bra Condition (Condition)	*p* = Sig.		
			T1	T2	T3
O	A	B	0.906	0.844	0.630
		C	0.542	0.085	0.808
		D	0.168	0.119	0.018 *
I	A	B	0.946	0.662	0.541
		C	0.840	0.159	0.915
		D	0.185	0.111	0.018 *
II	A	B	0.000 *	0.772	0.055
		C	0.000 *	0.209	0.837
		D	0.000 *	0.863	0.056
III	A	B	0.012 *	0.074	0.163
		C	0.005 *	0.314	0.979
		D	0.002 *	0.644	0.521
IV	A	B	0.019 *	0.183	0.287
		C	0.004 *	0.833	0.982
		D	0.005 *	0.918	0.404

* Significance at the 0.05 level is highlighted.

**Table 5 ijerph-19-15765-t005:** Subjective sensation rating of pressure comfort and support sensation of 4 bra conditions.

		Thermal Comfort				Pressure Comfort				Support Sensation			
		Bra A	Bra B	Bra C	Bra D	Bra A	Bra B	Bra C	Bra D	Bra A	Bra B	Bra C	Bra D
Average		6.90	6.71	6.67	7.00	7.19	7.19	7.00	7.24	7.86	8.29	7.90	7.90
Stand deviation		1.70	1.55	1.49	1.41	1.57	1.36	1.84	1.67	1.74	1.27	1.73	1.67
Maximum		10	10	9	9	9	9	10	9	10	10	9	9
Minimum		3	4	4	4	3	5	3	3	3	6	2	2

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
