# Peer review of "Sports Bra Pressure: Effect on Body Skin Temperature and Wear Comfort"

_ijerph, 2022, doi:10.3390/ijerph192315765_

Round 1

Reviewer 1 Report

First, congratulations to the authors for their effort and interest in this relevant topic. The following are my observations.

1.- Introduction.

Please better support this statement or be more specific “However, sports bras which exert high compressive forces on the chest may initiate other physiological and psychological effects on the body as the parameters for wear comfort have not been verified and pressure evaluations of bras have not been done in the literature”. For example, I indicate reference:

https://www.sciencedirect.com/science/article/abs/pii/S2352492815300581?casa_token=tsILNzvgwl8AAAAA:umXUCmqL7e-puEWibkwk2h4Z__tlPM93sLPewGJWsvXwxdAVg3SVCt46RgPPGeRXO-hWzIkZKY8C

2.- Materials and Methods

Use abbreviations of "minutes" in the singular "min" throughout the text.

Just as section 2.4 is established for the variable "Subjective Sensation Rating", for greater clarity of the protocol, establish a specific section for the variable "Skin temperature".

In Table 2, "Lowest thermal comfort" appears in bold. Please correct.

3.- Results

In table 3 "Underband" appears in bold. Please correct.

In Figure 5 "(a)" is missing.

In Figure 7 correct "(a)".

In Table 4 "Stage" and "Bra condition(base)" appear in bold. Please correct. Also, the "*" are not indicated (highlighted in yellow).

Table 5, Bra A to C is highlighted in bold. Please correct. Correct the abbreviation "Stand devia-tion"

 4.- Discussion.

Link references according to the adecuate format "[31][32]", [18][34]. Revise throughout the text.

Reviewer 2 Report

The paper is generally well written. I've few comments.

Abstract

1. Expand the results section with more details (i.e. results).

2. Add significance of finding at the end of the paragraph (take home message)

Introduction

3. However, sports bras which exert high compressive forces on the chest may initiate other physiological and psychological effects on the body as the parameters for wear comfort have not been verified and pressure evaluations of bras have not been done in the literature. Add citations for physiological and psychological affects.

4. More systematic research that investigates the pres-sure induced by sports bras and the associated impact on the body is therefore necessary to provide reliable and precise information which can be used to improve sports bra de-signs so that women can engage in more physical activities. This sentence is vague and needs to be revised (be more specific).

5. Nevertheless, excessive local skin pressure can harm the skin and hinder body movement, which are also considered harmful to the body because it leads to chronic constipation and diarrhoea, headaches and even visceral displacement... Suggest add citation here. 

6. Nevertheless, the mean pressure that is imposed onto the body from a bra greatly varies with anatomic location, body fat, body movement and posture which lead to different tactile and pressure sensations. Suggest add citations.

7. Add sysnthesis of findings at the end of paragraph two of the introduction.

8. I noticed that the word "discomfort" and "thermal discomfort" are used interchangeably throughout the manuscript. This is confusing. I suggest only use the term "thermal discomfort" throughout the manuscript.

9. Add sysnthesis of findings at the end of paragraph three of the introduction.

Methods

2.3 Experimental protocol

10. Add information of calibration/validation of the device.

Discussion

11. Paragraph one. On the other hand, to avoid breast injury,.... Perhaps discussed common types of injury caused by bra pressure.

Reference

Ref #7 typo, capital "I" for international.

Ref #10 Doi is added, not consistent with the rest.

Ref #27 typo in capitalization for journal name.

Ref #28 Doi is added, not consistent with the rest. 

Ref #34. Wrong information, See the link; https://www.jstage.jst.go.jp/article/transjtmsj1972/54/2/54_2_57/_article
